# Validation of Variant Assembly Using HAPHPIPE with Next-Generation Sequence Data from Viruses

**DOI:** 10.3390/v12070758

**Published:** 2020-07-14

**Authors:** Keylie M. Gibson, Margaret C. Steiner, Uzma Rentia, Matthew L. Bendall, Marcos Pérez-Losada, Keith A. Crandall

**Affiliations:** 1Computational Biology Institute, Milken Institute School of Public Health, The George Washington University, Washington, DC 20052, USA; steinerm@gwmail.gwu.edu (M.C.S.); uzma_rentia@gwmail.gwu.edu (U.R.); mlb4001@med.cornell.edu (M.L.B.); mlosada@gwu.edu (M.P.-L.); kcrandall@gwu.edu (K.A.C.); 2Department of Biostatistics and Bioinformatics, Milken Institute School of Public Health, The George Washington University, Washington, DC 20052, USA; 3CIBIO-InBIO, Centro de Investigação em Biodiversidade e Recursos Genéticos, Universidade do Porto, Campus Agrário de Vairão, 4169-007 Vairão, Portugal

**Keywords:** bioinformatics, validation, simulation, viruses, consensus, haplotypes, HIV, HCV, SARS-CoV-2

## Abstract

Next-generation sequencing (NGS) offers a powerful opportunity to identify low-abundance, intra-host viral sequence variants, yet the focus of many bioinformatic tools on consensus sequence construction has precluded a thorough analysis of intra-host diversity. To take full advantage of the resolution of NGS data, we developed HAplotype PHylodynamics PIPEline (HAPHPIPE), an open-source tool for the de novo and reference-based assembly of viral NGS data, with both consensus sequence assembly and a focus on the quantification of intra-host variation through haplotype reconstruction. We validate and compare the consensus sequence assembly methods of HAPHPIPE to those of two alternative software packages, HyDRA and Geneious, using simulated HIV and empirical HIV, HCV, and SARS-CoV-2 datasets. Our validation methods included read mapping, genetic distance, and genetic diversity metrics. In simulated NGS data, HAPHPIPE generated *pol* consensus sequences significantly closer to the true consensus sequence than those produced by HyDRA and Geneious and performed comparably to Geneious for HIV *gp120* sequences. Furthermore, using empirical data from multiple viruses, we demonstrate that HAPHPIPE can analyze larger sequence datasets due to its greater computational speed. Therefore, we contend that HAPHPIPE provides a more user-friendly platform for users with and without bioinformatics experience to implement current best practices for viral NGS assembly than other currently available options.

## 1. Introduction

Next-generation sequence (NGS) data provide a new opportunity to more efficiently study viral diversity, especially within-host sequence variation, which is key to understanding the evolutionary dynamics of viral populations both within and amongst hosts. NGS provides an opportunity to better explore viral sequence evolution over time [1] and evolution among hosts, including the direction of cross-species transmission [2], or elucidate the origin of viral epidemics [3]. While some studies capitalize on the ability of NGS data to identify intra-host sequence variants, the majority rely on consensus sequence estimation. This results in a loss of resolution in intra-patient viral diversity, which has nontrivial implications for downstream evolutionary inferences [4]. Thus, improving consensus sequence estimation methods is of great interest to the virology community.

There are two general approaches when constructing a consensus sequence from NGS data: de novo assembly and reference-based assembly (for reviews, see [5,6]). For reference-based assembly, sequencing reads are aligned (or mapped) to a reference sequence and a consensus sequence is then generated, often using majority rule, where the most frequently encountered nucleotide at each aligned position is chosen to be the nucleotide in the consensus sequence at that same position. Alternatively, consensus sequences can be generated using specified percentage cutoffs or by inserting ambiguity codes at sites with incongruities. De novo assembly does not require a reference sequence, but instead attempts to reconstruct the full sequence (or region of interest, such as an amplicon) by identifying overlapping nucleotides among the sequence reads. While reference-based assembly requires less memory, computational effort and sequencing depth, the generated consensus sequence may reflect the nucleotide composition of the reference sequence (i.e., bias towards the reference sequence) [7,8,9], thus potentially impacting the accuracy of downstream analyses. Issues in de novo assembly commonly arise from the large amount of computing effort required and the computational complexity of identifying overlapping regions in short reads. This issue is further compounded by highly variable short reads seen in quickly evolving retroviruses due to high genetic diversity, although small genomes still require relatively minimal computational power for assembly.

Recently, Ji et al. [10] recommended best practices for processing HIV NGS data, which include reference-based assembly using Bowtie2 [11] as the short read aligner and HXB2 (NCBI accession: K03455; [12]) as the reference sequence for constructing a consensus sequence. Many studies implement reference-based assembly [10,13,14] with tools such as CLC Main Workbench (Qiagen, Hilden, Germany) [15,16,17,18], Geneious (https://www.geneious.com) [19,20,21,22,23,24], HyDRA [25,26,27,28], SmartGene (Switzerland) [21,29,30], PAseq [31,32] and Amplicon Variant Analyzer (AVA; pyrosequencing-based platform) [21,33,34,35,36,37]. Other studies complete de novo assembly with tools such as Geneious [38], CLC Main Workbench (Qiagen) [16,39,40], and Iterative Virus Assembler (IVA) [22,41,42,43,44,45,46,47]. HCV studies follow similar patterns to those of HIV-1 [48,49,50,51,52,53,54,55,56,57,58,59]. A combination of both assembly approaches has been implemented to construct a consensus sequence by first mapping the reads to a reference sequence and then completing de novo assembly of those mapped reads [53,60].

Our software, HAplotype PHylodynamics PIPEline (HAPHPIPE), was designed to make viral NGS analyses more accessible and versatile for researchers and to provide the opportunity for identifying within-host variation by assembling variants from NGS data [61]. HAPHPIPE implements de novo or reference-based assembly followed by iterative refinement steps to assemble a better-representative consensus sequence for the entire viral population being surveyed. HAPHPIPE also implements haplotype reconstruction tools to facilitate the use of haplotypes in downstream analyses. The inclusion of haplotype data helps researchers to quantify within-host variation and, thereby, make improved inferences about associations with phenotypic traits.

A fundamental component of good software development is testing and validation [62]. Accordingly, we aim to test and validate HAPHPIPE using simulated HIV-1 data and empirical HIV-1 [63], HCV [64], and SARS-CoV-2 data. We hypothesize that in our simulation study: (i) due to the high genetic diversity of HIV-1, the de novo assembly strategy, regardless of platform, will produce a consensus sequence that is more genetically similar to the true sample isolate sequence than the reference-based assembly strategy; and (ii) that HAPHPIPE will perform as well as or better than HyDRA and Geneious, based on the metrics of read mapping, genetic distance, and genetic diversity. We also evaluate genetic distance to address potential bias in reference-based assembly. For the purposes of this study, we focus on the composition of the consensus sequences, but additionally report haplotype data for the HIV and HCV empirical datasets as validation of HAPHPIPE’s intra-host analytical capabilities. See Eliseev et al. [65] for an evaluation of haplotype reconstruction tools.

One aspect of note in NGS assembly specific to HIV is that often, these analyses are performed by users whose primary work is not in bioinformatics, such as clinicians, due to the often highly translational goals of HIV research. Because viral sequence analyses are rapidly becoming a staple of public health surveillance efforts, one of the primary goals of HAPHPIPE was to make intricate, command-based software accessible to introductory level bioinformatics users. Hence, we have included significant documentation and pre-scripted pipelines along with our software to facilitate its use by users of all backgrounds.

## 2. Materials and Methods

Below, we introduce the three tools that were used in generating consensus sequences, as well as the simulated and empirical datasets used in this study. Finally, we discuss the analyses completed for the pipelines and consensus sequences, along with true and reference sequences used (Figure 1). We compared HAPHPIPE to two other software programs, Geneious and HyDRA, based on their frequent use in viral studies [19,20,21,22,23,24,25,26,27,28,38,51,52,53,56,57], particularly among clinicians and those new to bioinformatics analysis. In particular, we chose HyDRA over similar web-based platforms such as PASeq [66] due to its popularity in the HIV research community; we chose Geneious as a representative of commercial software frequently used in genomics analysis. In selecting these methods, we target this validation study on performance for clinical and public health applications—which are especially pertinent for the empirical viral data included: HIV, HCV, and SARS-CoV-2. We suggest HAPHPIPE as a viable alternative to commercial and closed-source platforms for these applications.

### 2.1. HAPHPIPE

HAPHPIPE is a user-friendly tool designed for the customizable processing and analysis of viral NGS data [61]. HAPHPIPE is available for installation via Bioconda, a popular open-source, bioinformatics-specific software distribution system for Python packages. Installation of the HAPHPIPE suite requires only one command. HAPHPIPE is constructed in a modular format that has five main components: manipulating reads (*reads* stage), assembling consensus sequences (*assemble* stage), haplotype reconstruction (*haplotype* stage), post-analysis steps such as summary statistics or region extraction (*description* stage) and phylogenetics (*phylo* stage) [61]. There are two example pipelines included in the tool: haphpipe_assemble_01 and haphpipe_assemble_02 (Table 1). Haphpipe_assemble_01 is a de novo assembly pipeline that takes raw Illumina sequencing data, quality trims and error corrects the reads with Trimmomatic [67] and SPAdes [68], respectively, completes de novo assembly to form contigs with SPAdes, and forms scaffolds from the contigs with MUMMER 3+ [69]. Finally, the corrected reads are mapped back to the de novo assembled sequence with Bowtie2 [11]. The initial consensus sequence is updated through iterative refinement steps, where the corrected reads are repeatedly mapped back to the newly formed consensus sequence and the consensus sequence is updated with the new majority nucleotides at a position using a modified majority rule (taking quality scores and read depth into account). By default, this continues until the consensus sequence shows no improvement or changes to base composition or five refinement steps are completed. Alternatively, haphpipe_assemble_02 is a reference-based assembly approach, in which error corrected reads are mapped against a reference genome with Bowtie2 instead of being assembled de novo. All other steps are the same as in haphpipe_assemble_01.

HAPHPIPE also implements PredictHaplo [70] and CliqueSNV [71], two haplotype reconstruction tools. For purposes of this study, we present results generated by PredictHaplo, which was chosen for the HAPHPIPE suite because it was determined to have the best performance for capturing intra-host viral variation compared to eleven other haplotype reconstruction tools in a recent study [21] of diversity levels observed in viral intra-patient data. For a more thorough explanation of HAPHPIPE and detailed user instructions, see Bendall et al. [61] or https://gwcbi.github.io/haphpipe_docs/.

### 2.2. HyDRA

HyDRA is a freely available, web-based tool that utilizes a wrapper for Bowtie2 for reference-based mapping of Illumina MiSeq reads to HXB2, similar to the HAPHPIPE reference-based pipeline (Table 1). Briefly, the default parameters included: a minimum of 20% frequency for a base to be included in the consensus sequence, the default mutation database—which is the Stanford SDRM 2009 list of mutations—a target coverage of 10,000 reads, a minimum read length of 100 bp, a minimum average read quality score of 30, a sequencing platform error rate of 0.0021, a minimum variant quality of 30, a minimum read depth of 100 for a variant call, a minimum allele count of five to be considered a variant, and a minimum amino acid frequency of 0.01 for a mutation to be considered in the drug-resistant report.

Compared to HAPHPIPE, HyDRA requires a target read coverage (10,000 reads) whereas HAPHPIPE does not require a target. For quality trimming of the reads, the defaults for HAPHPIPE include trimming base pairs from the 3′ and 5′ ends of the reads that fall below a quality of 3 or contain ‘N’, removing any leftover adapter sequences, a sliding window of 4:15, which clips the read once the average quality of the window is below 15, and requires a minimum read length of 36 bp, which is less than HyDRA, which requires a minimum average quality score of 20 and a minimum read length of 100 bp. Rather than a set error rate, HAPHPIPE utilizes a more accurate correction tool with the built in error correction module of SPAdes, BayesHammer [72]. For variant calling, the default minimum variant quality is 15 in HAPHPIPE, again lower than HyDRA. Furthermore, HyDRA assembles data from each read pair separately, as opposed to pairing reads during assembly, and constructs one reconstructed *pol* region as opposed to multiple regions across the entire genome (i.e., *PRRT*, *int*, and *gp120*). HyDRA also does not allow for the assembly of envelope proteins and is restricted to only HIV-1 sequence analyses.

### 2.3. Geneious

Geneious is a commercial, desktop software that hosts a suite of bioinformatic tools to analyze sequence data (Table 1). For the purposes of this study, we followed the protocol detailed by Dudley et al. [19], which is a reference-based assembly. We paired the raw reads and then trimmed on both the 5′ and 3′ ends with an error probability limit of 0.001 using the modified-Mott algorithm (see Geneious documentation). We then mapped the trimmed reads against the reference sequence HXB2 with the Geneious mapper. Parameters were as follows: a maximum of 15 gaps were allowed per read, a maximum gap size of 15, a minimum overlap identify of 80%, a minimum word length of 14, an index word length of 12, a maximum of 15% mismatches per read, and a maximum ambiguity of 16, and searched more thoroughly for poor match reads. A consensus sequence was saved after mapping with a default base threshold of the highest quality, and for reads without a quality score at a particular base, a default threshold of 65% was used for the consensus sequence. This reference-based workflow in Geneious is similar to the HAPHPIPE reference-based pipeline but sets limits for gaps, does not include an error-correction step, and includes ambiguity codes in the consensus sequence. The default in HAPHPIPE uses the ‘fast-sensitive’ option in Bowtie2, which allows no mismatches in seed alignment (considerably lower compared to Geneious 15% mismatches per read), requires a seed length of 20 (larger than Geneious value of 12), and requires a seed interval of 1 + 0.50 using the square-root of the read length. For more detailed explanations of Bowtie2 parameters, see the user manual at http://bowtie-bio.sourceforge.net/bowtie2/manual.shtml.

We additionally adapted this reference-based workflow for de novo assembly. The same trimming parameters for quality control steps and the Geneious assembler were used for de novo assembly of contigs using the following parameters: variants with coverage over six were not merged, the merging of homopolymer variants, the production of scaffolds, a maximum of 15 gaps were allowed per read, a maximum gap size of 15, a minimum overlap identify of 80%, a minimum word length of 14, ignored words repeated more than 200 times, an index word length of 12, a reanalysis quality threshold of eight, a maximum of 15% mismatches per read, a maximum ambiguity of 16, and more thorough searching for poor match reads. Additionally, the de novo workflow required additional computational resources, so 32 GB of memory was allocated. The de novo assembly workflow, which is similar to the HAPHPIPE de novo pipeline, uses Geneious’s own proprietary de novo assembler to construct contigs, while HAPHPIPE uses SPAdes, which has been shown to produce longer and more accurately assembled contigs [68,73]. It is also fast and does not require as many computational resources as Geneious—requiring only 5 GB of memory for assembly and 8 GB of disk space. This is an important consideration for researchers wishing to conduct large-scale studies with many samples, as HAPHPIPE can also be run efficiently in parallel on an HPC cluster, whereas Geneious must be run in its GUI form. Lastly, in the Geneious de novo workflow, contigs had to be manually mapped back to reference sequences in order for amplicons to be identified, whereas in HAPHPIPE contigs are automatically scaffolded and labeled, thus reducing manual effort.

### 2.4. Data Simulation

A total of 100 HIV-1 subtype B genomes and 50 HIV-1 non-subtype B genomes were randomly pulled (with no duplicates) from the Los Alamos HIV Database (LANL; hiv.lanl.gov) reference genome 2017 list (Appendix A; [16,28,40,74,75,76,77,78,79,80,81,82,83,84,85,86,87,88,89,90,91,92,93,94,95,96,97,98,99,100,101,102,103,104,105,106,107,108,109,110,111,112,113,114,115,116,117,118,119,120,121,122,123,124,125,126,127,128,129,130,131,132,133,134,135,136]). For each sequence, all ambiguity codes (M, R, W, etc.) were replaced randomly with one of the corresponding nucleotides (M = A or C, R = A or G, etc.). We then extracted the protease and reverse transcriptase (*PRRT*, HXB2 numbering: 2252–3869), integrase (*int*, HXB2 numbering: 4230–5093), and *gp120* (HXB2 numbering: 6225–7757) gene regions (i.e., amplicons). Gaps were then removed from the each sequence, which was labeled as the “truesequence.fasta” for each sequence. We then simulated reads for each sequence based on the respective “truesequence.fasta” with ART v. MountRainier [137], a simulation tool that generates NGS reads from a consensus sequence. We simulated 150 bp paired-end reads with a 2000x fold coverage, a mean fragment size of 215 bp, and a standard deviation of 120 bp. These reads were also error-prone, implemented with the integrated Illumina MiSeq platform error profile, which means that the reads contained errors known to be caused by the sequencing platform itself, creating a realistic representation of a standard NGS dataset. This process resulted in a total of 25,000 paired-end reads per sequence, covering each of the three targeted gene regions. This procedure was repeated for all 100 subtype B and 50 non-subtype B sequences.

### 2.5. Analyses and Testing

The simulated FASTQ read files were used as the inputs for both assembly pipelines in HAPHPIPE (haphpipe_assemble_01 and haphpipe_assemble_02) [61]. HyDRA Web v. 1.5.1 and Geneious v. 10.2.6 were implemented using the workflows described previously, each of which resulted in a consensus sequence for each respective true sequence. For Geneious and HyDRA workflows, we additionally extracted amplicon regions by aligning to HXB2. We then aligned all consensus sequences generated from each pipeline and workflow using MAFFT v. 7.309 [138] and estimated genetic diversity with DnaSP v. 6 [139], specifically recording nucleotide diversity (π), Watterson’s theta (θ), variable sites, and estimated number of haplotypes. We did the same for the true sequences for purposes of comparison. The genetic distance for each generated consensus sequence with respect to the true sequence was calculated using proportional (p)-distances [140], the number of nucleotide differences per site over the length of the alignment. While consensus sequences constructed with HAPHPIPE did not contain ambiguity codes, those constructed by HyDRA and Geneious workflows contained many. As we wanted to account for these fairly, we also calculated an adjusted p-distance, which gave differences with ambiguity codes fractional weight. P-distances and adjusted p-distances were also calculated between each of the generated consensus sequences and the HXB2 reference sequence.

Results were visualized with R v. 3.6.0 [141] in R Studio v. 1.2.1335 [142] using the ggplot2 v. 3.1.1 [143] package. Hypothesis testing for genetic p-distance and adjusted p-distance results were performed using the non-parametric Kruskal–Wallis test [144,145] with the R package stats v 3.6.0 and the Dunn post-hoc test [146] v 1.3.5 for multiple comparisons in R. For the Dunn test, we report p-values adjusted with the Holm method [147]. Hypothesis testing for distance results between the initial and final consensus sequence generated by each HAPHPIPE pipeline was also performed using the Wilcoxon signed-rank test [148].

### 2.6. Empirical Data Applications

All SRA accessions, each representing empirical HIV-1 NGS amplicon data from two populations sampled repeatedly over one year, were selected from the BioProject accession PRJNA506879 [63]. The average read count per sample was 583,828 reads. We put the raw NGS reads for each of the 36 HIV samples through both HAPHPIPE pipelines, with the same amplicons used in the simulated HIV-1 data above and the HXB2 reference sequence. We generated haplotypes with PredictHaplo. We also processed the raw reads through HyDRA and Geneious in the same manner as the simulated HIV-1 data. We estimated the genetic diversity of the consensus sequences of the HAPHPIPE pipelines, HyDRA and Geneious with DnaSP. Finally, we calculated p-distances and adjusted genetic p-distances between a sequence (predicted haplotypes from both HAPHPIPE pipelines and the consensus sequences from both HAPHPIPE pipelines, HyDRA, and Geneious) and HXB2 reference sequence, using the same analysis steps as described above with the simulated data. Subtyping of all consensus sequences and haplotypes was performed using the REGA HIV subtyping tool [149].

A total of 23 SRA accessions of HCV sequence data were selected from Babcock et al. [64] (accession numbers: SRR1170557, SRR1170560-SRR1170568, SRR1170576-SRR1170579, SRR1170671-SRR1170679), each representing HCV viral variants from a patient cohort at several time points spanning two months. The average read count was 7.9 million paired-end reads per sample, and we analyzed these samples through the same pipelines as the empirical HIV-1 data above, except HyDRA, which is HIV-1-specific. For reference-based pipelines, we used the H77 HCV reference sequence (accession: NC_004102), which is of HCV subtype 1a [150]. The following amplicons were included: *core* (H77 numbering: 342–914), *E1* (H77 numbering: 915–1490), and *E2* (H77 numbering: 1491–2579). We also reconstructed haplotypes for this dataset, estimated genetic diversity for consensus sequences from each pipeline, and calculated the corresponding genetic distances (p-distance and adjusted p-distance) between each of the resulting pipeline consensus sequences and H77. Subtyping of all consensus sequences and haplotypes was performed using the Genome Detective HCV subtyping tool (www.genomedetective.com/app/typingtool/hcv/).

At the time of this study, four high-quality SARS-CoV-2 SRA samples were available (accession numbers: SRR11140744, SRR11140746, SRR11140748, SRR11140750), and we analyzed each through both HAPHPIPE pipelines and Geneious workflows. The average read count was 395,407 paired-end reads for the first three samples, while the last sample was smaller with only 17,208 reads. We used the severe acute respiratory syndrome coronavirus 2 isolate Wuhan-Hu-1 (accession NC_045512) as the reference sequence [151]. We sought to test whole genome assembly, so the entire genome was used (i.e., no amplicons were assembled for this dataset); this meant that, for the de novo assembly pipeline for HAPHPIPE (haphpipe_assemble_01), we used the numbering 0 to 29,902 for the reference GTF file. Again, we analyzed assembly statistics, genetic distance from reference metrics, and diversity estimates, as with the previously described datasets.

## 3. Results

For simplicity, we present and discuss results from adjusted p-distances, and when results from non-adjusted p-distances differ significantly, we addressed these inconsistencies within each section. All genetic p-distance results and the associated figures can be found in Appendix A.

### 3.1. Simulated Data

In the simulated data, the HAPHPIPE de novo pipeline did not proceed to a third refinement step (Table 2), which indicates that refining a second time did not improve the assembly. The reference-based pipeline, 93% and 88% of the simulated subtype B and non-subtype B samples, respectively, terminated at a third refinement step and only 2% and 4%, respectively, required a fourth refinement step (Table 2). The HAPHPIPE de novo pipeline produced >96% alignment rates, while refinement during the reference-based pipeline improved alignment rates further by an average of 6.72–21.1% (Table 2). Geneious produced lower mapping rates—82.63% and 63.05% for reference-based, and 64.20% and 64.30% for de novo workflows for subtype B and non-subtype B samples, respectively (Table 2). In subtype B samples, there was no significant difference in genetic distance from the true sequence between initial and final steps of the de novo pipeline (*p* value in the range [0.371, 1]) and distance from the true sequence decreased significantly after refinement for all genes except *int* in the reference-based pipeline (*p* < 0.001; Figure 2, Table A1). For non-subtype B samples, genetic distance from the true sequence between the initial and final steps decreased significantly for all genes except *int* (for which sequences were already extremely close to the true sequence) in the de novo pipeline (*p* < 0.001), and increased significantly in the reference-based pipeline for all genes (*p* < 0.001; Figure 2, Table A1).

The differences in the genetic distance among true and consensus sequences generated from all platforms were significant (*p* < 0.001, Appendix A), and post-hoc analysis indicated many significant pairwise comparisons (Figure 2, Table A2). Overall, for the simulation data, HAPHPIPE pipelines generated consensus sequences significantly closer to the true sequence than those produced by HyDRA (*p* < 0.001; Table A2) for all amplicons (*PRRT*, *int* and *pol*) in both subtype B and non-B sequences. Similarly, both HAPHPIPE pipelines generated consensus sequences significantly closer to the true sequence than Geneious reference-based workflow (all *p* < 0.001, subtype B for *gp120*: *p* < 0.05; Table A2). For subtype B sequences, HAPHPIPE pipelines generated consensus sequences significantly closer to the true sequence than those produced by both workflows in Geneious for *pol*, *PRRT*, and *int* (*p* < 0.001; Table A2). For non-subtype B sequences, HAPHPIPE generated consensus sequences significantly closer to the true sequence than those produced by the Geneious de novo workflow for *pol*, *PRRT*, and *int* (*p* < 0.001, except for Geneious de novo workflow vs. HAPHPIPE reference-based pipeline in *PRRT*: *p* < 0.05; Table A2). For both subtype B and non-subtype B sequences, in *gp120* for de novo assembly, there was no significant difference in distance from the true sequence between HAPHPIPE and Geneious consensus sequences (*p* = 0.427, 0.121; Figure 2, Table A2).

For subtype B sequences, there were no significant differences between distance to HXB2 before and after refinement in the HAPHPIPE de novo pipeline for all genes and in the reference-based pipeline for *pol* genes (*p* value in the range [0.167, 1]; Figure 3, Table A3). However, *gp120* reference-based consensus sequences were significantly closer to HXB2 post-refinement (*p* < 0.001; Table A3). In non-subtype B data, HAPHPIPE de novo consensus sequences were significantly further from HXB2 post-refinement across all genes (*p* < 0.01), however, reference-based *gp120* consensus sequences showed the opposite and were closer to HXB2 post-refinement (*p* < 0.001; Table A3). For *pol* genes, there were no significant differences in distance to HXB2 post-refinement in the reference-based consensus sequences (*p* value in the range [0.161, 1]; Table A3). Similar trends as those for the true sequence were seen in genetic distance to the HIV reference sequence, HXB2, for subtype B *pol* data; however, not for subtype B *gp120* or in general for non-subtype B data, which are less similar to HXB2 because of high variability and difference in subtype, respectively. Median distance to HXB2 was notably greater than distance to the true sequence in all genes for *gp120* sequences in the simulation dataset (Figure 3). Moreover, there were no significant differences between the genetic distance to HXB2 for consensus sequences from the HAPHPIPE pipelines and Geneious workflows for subtype B *gp120* sequences (*p* = 0.0799; Table A4, Appendix A). In non-subtype B data, there was no significant difference in the distance of *gp120* HAPHPIPE de novo and reference-based consensus sequences to HXB2 (*p* = 0.191; Table A4), while the same de novo consensus sequences were shown to be significantly closer to the true sequence (Table A2). Additionally, there was no significant difference in *PRRT* sequences between distance of the two HAPHPIPE consensus sequences (*p* = 0.774; Table A4) or between Geneious and HAPHPIPE reference-based sequences (*p* = 0.090) to HXB2 as there was in distance to the true sequence (Table A2). There was also no significant difference in *pol* in distance to HXB2 between the two HAPHPIPE pipelines (*p* = 0.879; Table A4), as was also seen in distance to the true sequence (Table A2).

As for genetic diversity, consensus sequences from HAPHPIPE (either pipeline) and Geneious reference-based workflow resulted in the greatest underestimations of nucleotide diversity (π) in *gp120*, which represents current diversity estimates [152], compared to estimates from the true sequence for both subtype B and non-subtype B sequences (Figure 4, Table A5). Reference-based assembly, in general, resulted in greater underestimations of π for non-subtype B sequences (Figure 4, Table A5). Similarly, all Geneious and HAPHPIPE consensus sequences resulted in underestimations of Watterson’s theta (θ), which represents historical diversity [152], for *gp120* in the non-subtype B sequences (Figure 4, Table A5). However, all of these differences were quite small, with a magnitude less than 0.08 for π and less than 0.04 for θ.

The results for adjusted p-distance differed from those of non-adjusted p-distance in some comparisons (Appendix A). Namely, using non-adjusted p-distance to the true sequence, we showed no significant difference between HAPHPIPE and Geneious reference-based sequences in *gp120* for both subtype B and non-B (Appendix A, Appendix A). In distance from HXB2 for subtype B sequences, we did find significant differences between methods (*p* < 0.001; Appendix A, Appendix A), in particular that HAPHPIPE and both Geneious workflows were closer to HXB2 than HAPHPIPE reference-based sequences and that HAPHPIPE and Geneious de novo sequences were closer to HXB2 than Geneious reference-based sequences (*p* < 0.05; Appendix A). In non-subtype B data, non-adjusted p-distance indicated that, in *gp120*, HAPHPIPE and Geneious de novo sequences were closer to HXB2 than HAPHPIPE reference-based sequences (*p* < 0.001), and that there was no difference between distance in Geneious and HAPHPIPE reference-based sequences (*p* = 0.541; Appendix A).

### 3.2. Empirical Data

Empirical HCV data were subsampled to 3 million reads per FASTQ file prior to assembly on all platforms due to memory limitations. For the Geneious de novo workflow, data were additionally subsampled to 100,000 reads per FASTQ file because at larger file sizes the assembly step failed to complete. Similarly, the empirical SARS-CoV-2 dataset was subsampled to 100,000 reads per FASTQ file for both Geneious workflows. However, both HAPHPIPE pipelines were able to run on the full dataset. No subsampling was necessary for the empirical HIV data on any platform.

In the HAPHPIPE pipelines, the majority of samples in all empirical data ceased at three refinement steps (Table 3). While the Geneious de novo alignment rates were much higher than those of both HAPHPIPE pipelines and the Geneious reference-based workflow for HIV and HCV data, these values reflect the percentage of reads mapped to contigs, which were then scaffolded to the reference sequence (Table 3). Only 27.78%, 29.07%, and 13.11% of these contigs mapped back to the reference for HIV, HCV, and Sars-CoV-2 data, respectively, and the number of reads included in the final scaffolded sequence is not reported. Contig mapping rates are not available for the HAPHPIPE pipelines because contigs are only used to build a scaffold—further refinement steps utilize reads directly instead of contigs. For the empirical HIV data, sequencing covered the entire genome as a set of five amplicons [63], while our assembly targeted only three genes as distinct amplicons. In the following subsections, we compare the effects of assembly methods among all platforms, as well as the implications of these assembly-related data, on the final consensus sequence.

#### 3.2.1. Empirical HIV Dataset

All consensus sequences and reconstructed haplotypes generated from the empirical HIV data were confirmed as Subtype B by the REGA subtyping tool. HAPHPIPE de novo sequences were significantly closer to HXB2 after refinement in all genes (*p* < 0.001; Figure 5, Table A6); while for HAPHPIPE reference-based sequences, only *gp120* was significantly closer to the reference genome (*p* = 0.003; Table A6). Consensus sequences produced for each gene in the empirical HIV dataset showed significant differences in genetic distance to the HIV subtype B reference sequence, HXB2 (*p* < 0.001; Table A7). Notably, HAPHPIPE reference-based consensus sequences were significantly closer to HXB2 than de novo consensus sequences across all gene regions (*p* < 0.001; Figure 5, Table A7); unlike in the simulated data, which showed similar results between both HAPHPIPE pipelines (Figure 3, Table A3). In fact, HAPHPIPE de novo sequences were significantly farther from HXB2 than any other pipeline in all three genes, despite the sample data also being subtype B. For Geneious, no significant differences were found between de novo and reference-based sequences in any gene (*p* value in the range [0.630, 1.00]; Table A7). There were also no significant differences found between any of the three reference-based pipelines in *pol* genes (*PRRT* and *int* (*p* value in the range [.251, 1.00]), yet in *gp120* HAPHPIPE, reference-based sequences were significantly closer to HXB2 than those from Geneious (Table A7). In HAPHPIPE, reference-based sequences for *PRRT* and *gp120*, reconstructed haplotypes were significantly farther from HXB2 than the HAPHPIPE reference-based consensus sequences (*p* < 0.05), as were de novo *gp120* sequences (*p* < 0.05). Haplotypes for de novo *PRRT* sequences and *int* sequences from both pipelines showed no significant difference in distance from HXB2 (Table A7). In all three gene regions, haplotypes had considerably more variable sites than consensus sequences (Table 4). Compared to consensus sequences, haplotypes also had higher values of Watterson’s theta in all genes and higher values of pi in *PRRT* and *gp120* (Table 4). HyDRA sequences had the lowest values of both pi and theta for both *pol* genes (Table 4). Both diversity metrics were also higher for HAPHPIPE pipelines than their Geneious counterparts, in both de novo and reference-based workflows (Table 4).

#### 3.2.2. Empirical HCV Dataset

Notably, subtyping results from the Genome Detective HCV subtyping tool indicate that 42.2% of consensus sequences could not be assigned to a subtype, including sequences from each sample and from each platform and assembly method. The remaining sequences were assigned to subtype 1a, the same as the H77 reference sequence. This indicates that the data are likely not purely type 1a and may include either significant mutations or recombination with other HCV subtypes. HAPHPIPE de novo sequences were significantly closer to H77 after refinement for all genes (*p* < 0.001), as were HAPHPIPE reference-based sequences for *core* and *E2* (*p* < 0.01), but no significant difference was seen in *E1* (*p* = 0.058; Figure 5, Table A8). For all three genes (*core*, *E1*, and *E2*), HAPHPIPE reference-based consensus sequences were closer to H77 than HAPHPIPE de novo consensus sequences (*p* < 0.001; Figure 5, Table A8). HAPHPIPE de novo sequences were significantly closer to H77 than Geneious de novo sequences in *core* and farther in *E1* (*p* < 0.01), though this difference was not significant in *E2* (*p* = 0.069; Figure 5, Table A8). Geneious de novo sequences were significantly farther from H77 than Geneious reference-based sequences in both *core* and *E2* (*p* < 0.05), but not in *E1* (*p* = 0.201, Figure 5, Table A8). HAPHPIPE reference-based consensus sequences were significantly closer to H77 than all other pipelines in *core* (*p* < 0.01) and showed no significant differences compared to either Geneious workflow in the envelope genes (*E1*, *E2*; *p* value in the range [0.369, 1]; Figure 5, Table A8). No significant differences in distance from H77 were seen between consensus sequences and their respective haplotypes for either HAPHPIPE pipeline in all genes (*p* value in the range [0.702,1.00], Table A8). The number of variable sites in haplotypes versus consensus sequences was highly variable, with no consistent trends seen across genes (Table 5). Overall, diversity estimates for Geneious de novo sequences were higher than the other constructed consensus sequences, which could be related to the extreme subsampling for this dataset (Table 5).

#### 3.2.3. Empirical SARS-CoV-2 Dataset

Genetic distance from the reference sequence, Wuhan-Hu-1, was low across all four samples and pipelines, ranging from 0.00097 to 0.10384 (Table 6). Geneious de novo consensus sequences, though, showed the highest average genetic distance from the Wuhan-Hu-1 reference sequence, as well as the highest standard deviation in distance, indicating that these results are uniquely far from the reference sequence—even when compared to HAPHPIPE de novo consensus sequences—and that the results are most variable, likely as a result of the extreme subsampling of reads for the Geneious platform (Table 6).

#### 3.2.4. Differences between Genetic Distance Measurements

There were few differences in results based on non-adjusted p-distance in the empirical data. Non-adjusted p-distance results indicate that for *PRRT*, both HAPHPIPE and Geneious reference-based consensus sequences were closer to HXB2 than those from HyDRA (*p* < 0.05); HAPHPIPE reference-based sequences were also closer to HXB2 than Geneious de novo sequences (Appendix A, Appendix A). Additionally, the difference between de novo *gp120* sequences and haplotypes was not significant (*p* = 0.057; Appendix A). In the HCV *core*, HAPHPIPE de novo sequences were closer to H77 than Geneious de novo (*p* < 0.05), while in *E1* both HAPHPIPE and Geneious reference-based sequences were closer to H77 than Geneious de novo (*p* < 0.001), and there was no difference between HAPHPIPE and Geneious de novo (*p* = 0.573). In *E2,* HAPHPIPE reference-based sequences were closer to H77 than Geneious de novo (*p* < 0.001; Appendix A).

## 4. Discussion

In this study, we benchmarked the performance of HAPHPIPE consensus sequence assembly with two commonly used alternatives, HyDRA and Geneious. We found that in the simulation study, HAPHPIPE performed better than HyDRA, and while it performed equally well or better than Geneious, it can, in addition, handle larger datasets with greater speed. We also validated the performance on HAPHPIPE with empirical data. In all analyses, we addressed genetic distance from true sequences and/or reference sequences, genetic diversity, and assembly statistics, and in the real data we additionally constructed haplotypes using HAPHPIPE’s PredictHaplo implementation. When we discuss genetic distance, we refer to adjusted genetic p-distance, which takes into account ambiguity codes, unless otherwise specified. Lastly, we place our software into context with other NGS assembly software.

### 4.1. Simulated Data

For the simulated dataset, mapping rates were higher in both HAPHPIPE pipelines than both Geneious workflows. We also found that often two refinement steps were efficient in creating a better, more representative consensus sequence. In subtype B simulated data, the iterative refinement step in HAPHPIPE resulted in final sequences that were closer to the true sequence than the initial consensus for most genes in reference-based assembly. In de novo assembly, initial consensus sequences were already very close to the true sequence before refinement, and so no significant difference in distance was seen after this step. In non-subtype B data, sequences in reference-based assembly were significantly closer to the true sequence after refinement, yet sequences in de novo assembly were significantly farther; this is likely due to the difference in subtype between the consensus and the reference. The use of subtype-specific references would be necessary to evaluate the extent that these differences impact performance.

De novo assembly with HAPHPIPE produced a consensus sequence that is significantly more genetically similar to the true sample isolate sequence compared to reference-based assembly with HyDRA or Geneious. This was especially notable in non-subtype B HIV-1 *pol* sequences, which are most commonly targeted for clinical applications. There was no significant difference in genetic distance for the simulated HIV datasets between the HAPHPIPE pipelines, except for the *gp120* amplicon, in which de novo assembly constructed a consensus sequence genetically closer to the true sequence. De novo assembly for *gp120* with Geneious and HAPHPIPE constructed similar consensus sequences and provided a more accurate representation of the true sequence overall compared to reference-based assembly pipelines. This suggests that for more diverse genes such as envelope proteins, using either tool’s de novo option would result in similarly constructed consensus sequences that are more likely to be accurate than reference-based counterparts. Adjusted and non-adjusted genetic p-distance results to the true sequence were not identical, with non-adjusted genetic p-distance having shown no significant difference between HAPHPIPE and Geneious reference-based sequences in *gp120* for both subtypes. We discuss genetic distance to HXB2 below in Section 4.3 (bias towards reference sequence).

### 4.2. Empirical Data

#### 4.2.1. Assembly Statistics and Performance

Two main aspects of assembly, which varied across platforms for both de novo and reference-based assembly, were feasible input data size and read mapping rates. Geneious workflows required extensive subsampling that HAPHPIPE pipelines did not. These subsampling steps undoubtedly affected assembly statistics and the resulting consensus sequences, and the inability of Geneious to complete assembly with higher than 100,000 reads limits its comparison to HAPHPIPE. For example, the higher read mapping rate of Geneious de novo assembly in the empirical HCV data may be artificially high compared to HAPHPIPE, as far fewer reads were available to begin with and less than a third of assembled contigs successfully scaffolded to the reference sequence.

Additionally, the empirical HIV data covered the entire genome as a set of five amplicons, while here we constructed only three. Thus, the relatively low mapping rates reflect that not all sequencing reads mapped to the three genes (*PRRT*, *int*, and *gp120*), and that the high mapping rate of reads in Geneious de novo assembly could indicate incorrect mapping of reads from elsewhere in the genome to these three gene regions. Similar to HCV, the mapping rates across platforms for HIV reflected high rates for Geneious compared to HAPHPIPE. While not an effect of subsampling, this result is limited by less than a third of assembled contigs scaffolding to the reference and may also indicate incorrect mapping of reads into amplicons. In SARS-CoV-2 assembly, where the entire genome was used and sequences differed much less from the reference, HAPHPIPE mapping rates were higher, supporting that these factors may have contributed to the opposite pattern seen in the other datasets.

It is also notable that mapping rates were lower after the refinement step for all empirical datasets. This effect is most pronounced in the HAPHPIPE de novo pipeline. However, in the simulation dataset, mapping rates either remained the same post-refinement or increased, particularly in the reference-based pipeline. It is possible that the decrease in mapping rates post-refinement in the empirical data is due to the initial incorrect mapping of reads to genome regions, in which case removing these reads at the refinement step helped to improve the specificity of the alignment. This effect would not be seen in the simulated data, as all reads were derived from the specific amplicons used. Further, the increase in the mapping rates of simulated data for the reference-based pipeline may be due to incorrect initial mapping to the reference sequence during assembly, which was then corrected in the refinement step, leading to a higher overall mapping rate.

#### 4.2.2. HIV

In the empirical HIV data, HAPHPIPE de novo consensus sequences were consistently farther from the reference sequence, HXB2, than all others, including HyDRA. In HAPHPIPE, the refinement of de novo sequences decreased distance from HXB2, while in reference-based sequences this was only seen in *gp120*. In general, the reconstructed haplotypes were significantly farther from HXB2 than their respective HAPHPIPE consensus sequences only in *gp120* for the de novo pipeline, and in both *PRRT* and *gp120* for the reference-based pipeline. Haplotypes from *PRRT* and *gp120* in both pipelines also showed increased genetic diversity in three metrics: variable sites, pi, and Watterson’s theta. The observation that the haplotypes exhibited more diversity and greater distance from HXB2 than reference-based consensus sequences suggests that these consensus sequences may have been biased towards HXB2. While all consensus sequences were confirmed to be of subtype B, divergence from HXB2 could likely be a result of drug resistance mutations. As such, the greater distance of HAPHPIPE de novo sequences from HXB2 as compared to consensus sequences from other methods may reflect higher accuracy to the “true” sequence, although this hypothesis cannot be directly tested, highlighting a key limitation in the analysis of empirical data for this purpose.

#### 4.2.3. HCV

Results from the empirical HCV data were considerably more variable than those from the empirical HIV data. Pairwise differences in distance from H77 across platforms yielded inconsistent results, particularly in envelope genes (*E1* and *E2*). In contrast to the empirical HIV dataset, haplotypes were not significantly more or less distant from H77 than the respective HAPHPIPE consensus sequences in any gene. While haplotypes again showed slightly higher metrics of genetic diversity, overall these metrics were highly variable. Additionally, the Geneious reference-based workflow showed the highest pi and Watterson’s theta for the *core* gene by an order of magnitude larger compared to the others, including haplotypes. Two main confounders in these data are subsampling of Geneious de novo data to 100,000 reads and inconsistency in subtype, with over 40% of consensus sequences obtained—from each sample and each method—being of indeterminate type. This limits the use of H77 as an appropriate standard of comparison. Thus, more definitive conclusions from the empirical HCV data cannot be made.

#### 4.2.4. SARS-CoV-2

For both HAPHPIPE and Geneious platforms, the distance of assembled SARS-CoV-2 sequences from the reference, Wuhan-Hu-1, was quite low. Geneious de novo consensus sequences were the most variable in distance from the reference and, on average, were farther from the reference. This effect was not seen in HAPHPIPE de novo sequences, implying that it may be due to the subsampling of SARS-CoV-2 data to 100,000 reads for Geneious. A greater number of samples would be necessary to investigate this claim, however at the time of this study limited NGS data were publicly available, thus limiting our analysis and potentially contributing to the low distance to reference.

#### 4.2.5. Effects of Ambiguity Codes

Adjusted and non-adjusted genetic p-distance results, with respect to the reference sequences for the empirical datasets, were not identical. In particular, in HIV data, HAPHPIPE and Geneious showed significant differences when using non-adjusted p-distance, but these differences were not significant when using adjusted p-distance. In HCV data, most notable differences occurred in pairwise comparisons involving the Geneious de novo workflow, in which these consensus sequences were further from the reference sequence with non-adjusted p-distance. This observation is not surprising given the increase in ambiguity codes in Geneious de novo workflow, again likely due to the extreme subsampling. In SARS-CoV-2 data, there were no differences in non-adjusted p-distance and adjusted p-distance for HAPHPIPE consensus sequences, while in both Geneious workflows, adjusted p-distance was higher, again due to the inclusion of ambiguity codes in Geneious consensus sequences, likely compounded with the subsampling as well.

### 4.3. Bias towards Reference Sequence

In this study, we used HXB2 consistently for all HIV subtypes for purposes of comparison with HyDRA, which only uses HXB2 and does not allow the user to change the reference sequence. The refinement of HAPHPIPE de novo consensus sequences resulted in greater distance from HXB2 in the non-subtype B simulated dataset. Reference-based sequences in *gp120* were closer to HXB2 post-refinement in both datasets. These results suggest potential bias of non-subtype B *pol* sequences and *gp120* sequences—regardless of subtype—to HXB2 in reference-based assembly. Moreover, distances from HXB2 were significantly different to distances from the true sequence in the *gp120* region of subtype B sequences and in all genes except *int* in non-subtype B sequences. This observation suggests that the use of HXB2 as the reference introduced bias to the consensus sequences in variable regions of subtype B sequences (i.e., *gp120*) and in multiple regions of non-subtype B sequences. Placed in this context, it is likely that bias towards HXB2 would be most present in variable regions, regardless of subtype, and that the inconclusive subtype of the empirical HCV data may have introduced nontrivial bias to results. These results further suggest that in practice, subtype-specific reference sequences should be used whenever possible. Although this change in reference is possible with some viral assembly pipelines, it is a key feature that cannot be changed in HyDRA, which is often used in clinical practice to identify drug-resistant mutations. Thus, we support the conclusion that de novo assembly may be better than reference-based assembly for variable regions of the genome (e.g., envelope genes) and when subtype is either mixed or uncertain.

### 4.4. Limitations

Our study has a few notable limitations. First, we were primarily constrained by the lack of knowledge of “true” sequence compositions for the empirical data. Therefore, we were unable to determine how genetically similar the constructed consensus sequences were to the “true” sequence(s) for each pipeline. However, consensus sequences do not occur in the viral population thus distinguishing them from ancestral sequences [119,153]; therefore, obtaining a “true” consensus sequence is not entirely attainable. Still, consensus sequences are often used in a case of best representative sequence for computational efficiency or, often in clinical applications, for the identification of drug-resistant mutations to guide medical treatment. As such, even if presented with a “true” consensus sequence for comparison, as we have done in the data simulation section, we still compare to an estimated sequence that may or may not reflect true intra-patient diversity, thus limiting the generalization of resulting conclusions. NGS circumvents this limitation by facilitating the reconstruction of viral haplotypes, which have been shown to improve the resolution of phylodynamic inferences [154]. Thus, the implementation of haplotype reconstruction methods in HAPHPIPE facilitates more accurate, informative analyses of both intra- and inter-patient viral diversity and evolution.

Another notable limitation we encountered was the inability of Geneious workflows to be completed on the full set of reads for each of the HCV and SARS-CoV-2 empirical data. Therefore, for both of these datasets, the resulting Geneious consensus sequences were likely skewed compared to the HAPHPIPE pipelines, which did not require as extensive subsampling for HCV data and required no subsampling for HIV or SARS-CoV-2 data. Furthermore, it took each empirical sample at least two days to complete the de novo assembly pipeline on Geneious. We hypothesize that Geneious had trouble with viral NGS data because viruses are fast-evolving and contain many variants, thus it was difficult for the program to orient the reads together for de novo assembly, as in HCV, or even against a reference, as in the case of SARS-CoV-2. We further hypothesize that the large size of both datasets may have posed memory issues during the run, as likely indicated by the increased time to completion, which could impact functionality. This caveat in Geneious further emphasizes the applicability of HAPHPIPE for assembling viral sequences from NGS data. At the time of this study, limited NGS data were available for SARS-CoV-2 sequences. In particular, only four samples of good quality were publicly available, thus limiting this aspect of our analysis.

### 4.5. Utility of Software

Geneious and HyDRA both present downsides in practical use. Although Geneious is an interactive software that is easy to use with its graphical user interface (GUI), it does require payment. It is also cumbersome and time consuming to complete assemblies, especially de novo assemblies, on large-scale projects with many NGS samples. We streamlined the process by making workflows, but in doing so, it was time-intensive to rename output files efficiently and distinguish output files. Furthermore, we had trouble, despite allocating ample memory and subsampling reads, with assembly workflows for the empirical datasets. Moreover, each sample processed through the de novo assembly workflow took at least two days to finish and produce a consensus sequence, with some empirical HIV samples taking upwards of seven days. While Geneious does include or have available extensions to phylogenetic tools such as multiple sequence alignment and building trees, the full capabilities of such software may not be available or feasible to be run locally, thus necessitating the use of an additional tool for phylodynamic steps.

Although HyDRA is a free, online-based software, it produced the least accurate results here for HIV-1 in our simulation study. This drop in accuracy could be due to HyDRA’s inability to analyze paired-end data together. Moreover, HyDRA only allows for the assembly of polymerase genes for HIV, due to the emphasis on drug-resistant mutations, and does not allow the user to change the reference sequence. Both Geneious and HyDRA constructed consensus sequences with many ambiguity codes, which could be due to intra-host variation or drug-resistant mutations. HyDRA does not include options for phylogenetics steps.

The main advantage of Geneious and HyDRA as compared to HAPHPIPE, particularly for those unfamiliar with UNIX-based command line and bash, is that the former two include an easy-to-use GUI. However, for more advanced users and those analyzing large-scale datasets, this becomes a hindrance to efficiency. Additionally, storing large-scale NGS datasets locally on the user’s computer as opposed to remotely on a high-performance cluster, may pose limitations for larger studies in using GUIs. While HAPHPIPE does require knowledge of the command line interface, the example pipelines given (haphpipe_assemble_01 and haphpipe_assemble_02), as well as the thorough documentation and beginner-focused user guide [61], simplify this process for non-bioinformatic users. We have also simplified the installation process for HAPHPIPE by adding it to Bioconda, a popular bioinformatics software repository. The command line interface of HAPHPIPE presents several clear advantages over the GUI-based programs used in this study, namely the ability to run on a high-performance cluster using parallelization techniques, compatibility with bash scripting to automate assembly of many samples at once, and extensibility both for custom pipelines and for additional modules. One further limitation of HAPHPIPE, specifically in clinical and public health applications, is the lack of a module for HIV drug-resistance identification. However, HAPHPIPE is applicable for a variety of viruses and does include general variant calling as well as consensus sequences in formats compatible with existing DRM identification tools, such as Stanford HIVdb [155,156]. HAPHPIPE provides an additional step in analyzing NGS data from intra-host populations by implementing wrappers for estimating haplotypes. By performing haplotype reconstruction, in addition to consensus assembly, our approach can output a more detailed representation of the haplotype diversity in a sample from NGS data. Therefore, HAPHPIPE can be used both to create a more accurate consensus sequence and to capture viral variants within the data by presenting reconstructed haplotype sequences.

### 4.6. Comparison to other Viral Pipelines

In this study, we compare the performance of HAPHPIPE to that of Geneious and HyDRA, both GUI-based tools that are most commonly used among clinicians and others whose primary background is not necessarily bioinformatics. However, several additional viral assembly pipelines on the command line exist and may be compared to HAPHPIPE. While a full comparison of the performance of these methods with respect to HAPHPIPE is outside the scope of this validation study, here we discuss three in this context: viral-ngs, MiCall, and V-pipe.

Viral-ngs is an open source, command-line package (https://github.com/broadinstitute/viral-ngs) that utilizes Trinity [157]. De novo assembly with SPAdes is offered as an alternative option. Following de novo assembly, Gap2Seq [158] is used to fill the gaps in the generated scaffold. The remaining steps in the assembly portion of viral-ngs uses reference-based assembly improvements to generate the final consensus sequence. Like HAPHPIPE, specifically the de novo pipeline, viral-ngs uses MUMMER to orient and merge contigs with the assistance of a reference FASTA sequence, however, HAPHPIPE additionally includes an option to construct amplicon-specific sequences at this stage. In the final, major step of viral-ngs assembly, viral-ngs uses Novoalign (http://www.novocraft.com)—which requires a commercial license—to call back reads and align them to the crude de novo assembly, which is then iteratively improved. In both HAPHPIPE and viral-ngs, this step is named refine_assembly and has essentially the same function, although HAPHPIPE allows the user to define the number of desired iterations (with default settings being set at five iterations), while the analogous stage in viral-ngs is set at two iterations. Viral-ngs also implements an additional imputation step as a part of refinement. Lastly, HAPHPIPE and viral-ngs include distinct focuses for downstream analysis, namely phylodynamics versus metagenomics, respectively, and thus may appeal to different user bases.

The reference-based assembly pipeline MiCall (https://github.com/cfe-lab/MiCall) is based on Bowtie2—as are viral-ngs and HAPHPIPE—and has been noted in the literature to be interchangeable with platforms such as HyDRA and PASeq. Like HAPHPIPE, MiCall includes both assembly options. Following quality control measures, the reads are mapped to a reference amplicon. From there, reads are either assembled in relation to a reference or used to create contigs de novo. Then in a stage akin to HAPHPIPE refine_assembly, reads are mapped onto the previously created consensus and improved. Unlike HAPHPIPE, however, this stage is only executed iteratively for reference-based assemblies. As a pipeline tailored to HIV, like HyDRA, MiCall also has a stage, ‘resistance‘, that determines the reads’ resistance to antiretroviral therapies (ART). MiCall is more HIV and HCV specific, whereas HAPHPIPE has more applicability across many other viral species.

V-pipe is a publicly available, command line tool (https://cbg-ethz.github.io/V-pipe/) in which reads are mapped to an initial reference to generate a crude alignment. The initial reference can be provided or created de novo from the software VICUNA [159]. The pipeline then uses a Hidden-Markov Model (HMM)-based aligner designed for NGS reads of small genomes that are prone to indels, such as HCV and HIV, (ngshmmalign; https://github.com/cbg-ethz/ngshmmalign). The original reads are then mapped against the profile HMM. The creation of the HMM profile is unique to V-pipe, as is its utilization of VICUNA and ngshmmalign over the established BWA [160] and Bowtie2 wrappers. Like HAPHPIPE, but unlike viral-ngs and MiCall, V-pipe calls haplotypes. Rather than using PredictHaplo, V-pipe implements HaploClique and Savage for global haplotype reconstruction and ShoRAH for local reconstruction, all of which performed poorly relative to PredictHaplo in a recent comparison of haplotype reconstruction tools [65].

## 5. Conclusions

We found that NGS viral analysis is improved with the use of HAPHPIPE, particularly in conserved regions. Furthermore, we demonstrated that de novo assembly performs better than reference-based assembly at generating a consensus sequence that is closer to the true sequence. Additionally, we further validated the performance of HAPHPIPE across multiple viruses of varying genome lengths, as well as both amplicon and whole genome viral assembly from NGS data. We found that HAPHPIPE facilitated the use of a greater quantity of empirical data and completed assemblies more quickly than other methods, in particular for datasets of viruses with greater genomes (e.g., SARS-CoV-2 whole-genome assembly) and with greater sequencing depth (e.g., the empirical HCV data used here). While in this study we mainly focused on two commonly used GUI-based tools, we also compared the functionality of our software to other available command-line platforms. A thorough comparative study of the performance of the many terminal-based viral assembly tools and pipelines would be of great value to the research community.

Based on the conclusions of this validation study, we believe that HAPHPIPE provides a more efficient and informative pipeline for the analysis of NGS viral data, particularly for translational clinical and public health research. HAPHPIPE is a single, open-source tool that allows for customization of the analyses, generates a more accurate viral consensus sequence, and produces properly formatted outputs for further phylodynamic analyses, as well as integrates these methods into a unified framework. By including user-friendly wrappers for complex bioinformatics programs, detailed documentation, and a beginner-level User Guide and protocol publication, as well as by maintaining our program as open-source, freely available software, we expect HAPHPIPE to make sophisticated genomics analysis more accessible to researchers across many biomedical fields.

## Figures and Tables

**Figure 1 viruses-12-00758-f001:**
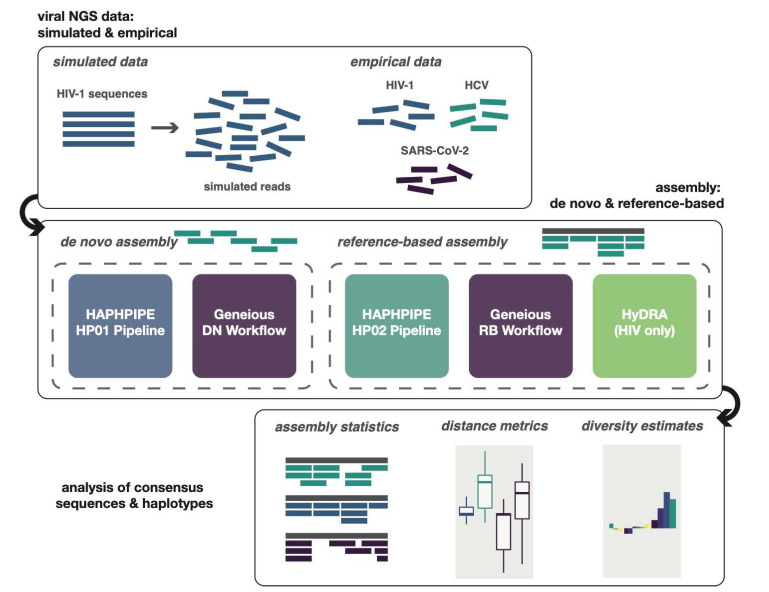
Methods overview. Sequencing reads were simulated for each simulation sample, while reads for empirical data were gathered from NCBI SRA database. Reads for each sample were assembled and a consensus sequence generated through the de novo pipelines for HAPHPIPE (HP01) and Geneious and the reference-based pipelines for HAPHPIPE (HP02), Geneious, and HyDRA. Only HIV samples were analyzed through HyDRA, because HyDRA is HIV-specific. All resulting consensus sequences were analyzed using a variety of metrics including assembly statistics, genetic distance from reference or true sequence metrics, and diversity estimates, such as nucleotide diversity.

**Figure 2 viruses-12-00758-f002:**
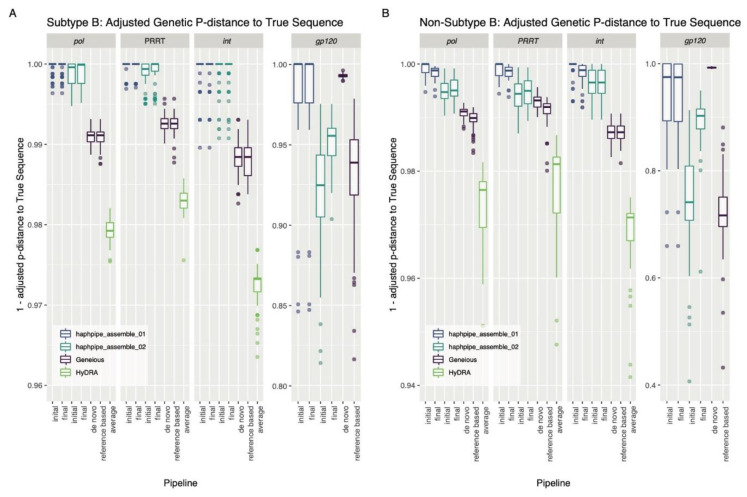
Adjusted genetic p-distance (displayed as a difference from 1) between consensus sequence and true sequence for all pipelines for the simulated HIV (**A**) subtype B dataset and (**B**) non-subtype B dataset. Ambiguous nucleotides were accounted for by giving fractional weight in alignment. A value closer to 1.00 indicates the consensus sequence is more genetically similar to the true sequence. The *x*-axis order from left to right for an individual panel: adjusted genetic p-distance between the true sequence and (i) the initial assembled sequence followed by (ii) the final assemble sequence for haphpipe_assemble_01 pipeline (de novo assembly); (iii) the initial assembled sequence followed by (iv) the final assemble sequence for haphpipe_assemble_02 pipeline (reference-based assembly); the final consensus sequence for the Geneious (v) de novo workflow and the (vi) reference-based workflow; and finally, the (vi) average between the final two sequences (one for each read file) for HyDRA. The three amplicons are shown, as well as a combination of *PRRT* and *int* amplicons into *pol*. There are no results for HyDRA in the *gp120* gene because HyDRA only analyzes the *pol* gene.

**Figure 3 viruses-12-00758-f003:**
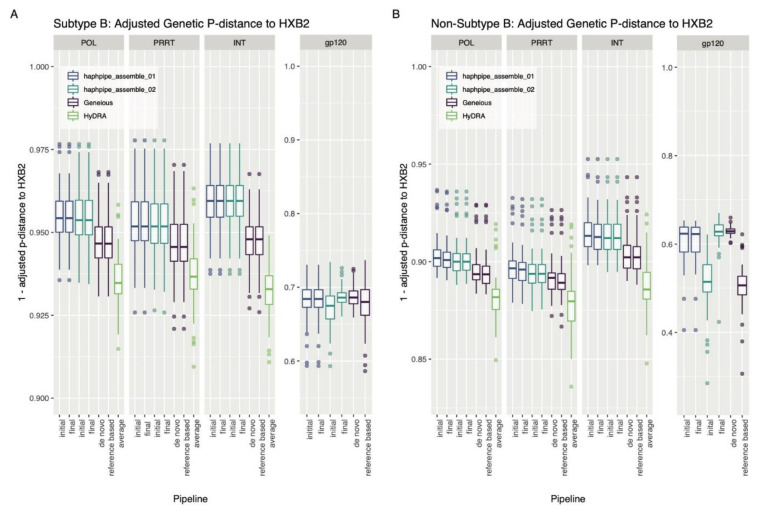
Adjusted genetic p-distance (displayed as a difference from 1) between consensus sequence and HXB2, the reference sequence for HIV, for all pipelines for the simulated HIV (**A**) subtype B dataset and (**B**) non-subtype B dataset. Ambiguous nucleotides were accounted for by giving fractional weight in alignment. A value closer to 1.00 indicates that the consensus sequence is more genetically similar to the reference sequence. The *x*-axis order from left to right for an individual panel: adjusted genetic p-distance between the reference sequence and (i) the initial assembled sequence followed by (ii) the final assemble sequence for haphpipe_assemble_01 pipeline (de novo assembly); (iii) the initial assembled sequence followed by (iv) the final assemble sequence for haphpipe_assemble_02 pipeline (reference-based assembly); the final consensus sequence for the Geneious (v) de novo workflow and the (vi) reference-based workflow; and finally, the (vi) average between the final two sequences (one for each read file) for HyDRA. The three amplicons are shown, as well as a combination of *PRRT* and *int* amplicons into *pol*. There are no results for HyDRA in the *gp120* gene because HyDRA only analyzes the *pol* gene.

**Figure 4 viruses-12-00758-f004:**
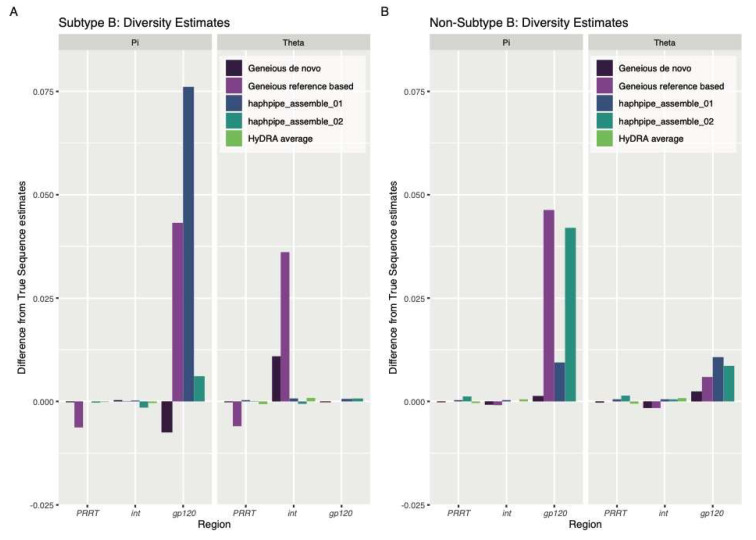
Difference between the estimated genetic diversity from the true sequence and each pipeline (calculated as estimate of true sequences—estimate of pipeline consensus sequences) for the simulated HIV (**A**) subtype B dataset and (**B**) non-subtype B dataset. Positive value indicates an underestimation of the genetic diversity with the consensus sequences from the pipeline, and a negative value indicates an overestimation of the genetic diversity with the consensus sequences from the pipeline. *PRRT* = protease and reverse transcriptase, *int* = integrase, *gp120* = gene within envelope gene region, Pi = nucleotide diversity, Theta = Watterson’s genetic diversity.

**Figure 5 viruses-12-00758-f005:**
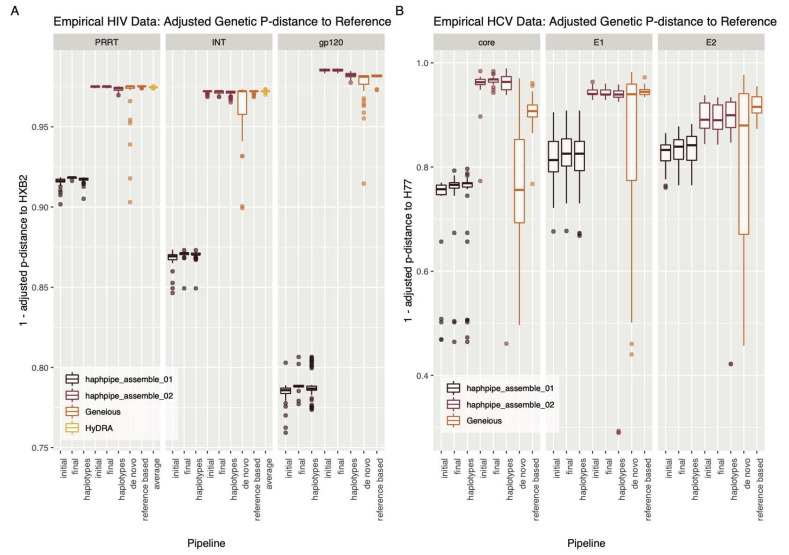
Adjusted genetic p-distance (displayed as a difference from 1) between consensus sequence and HXB2, the reference sequence for HIV, for all pipelines for the empirical (**A**) HIV dataset and (**B**) HCV dataset. Ambiguous nucleotides were accounted for by giving fractional weight in alignment. A value closer to 1.00 indicates that the consensus sequence is more genetically similar to the reference sequence. The y-axes are different for each HIV and HCV, with HCV showing greater variance between samples. The *x*-axis order from left to right for an individual panel: adjusted genetic p-distance between the reference sequence and (i) the initial assembled sequence, (ii) the final assemble sequence and (iii) the reconstructed haplotypes for haphpipe_assemble_01 pipeline (de novo assembly); (iv) the initial assembled sequence, (v) the final assemble sequence, and (vi) the reconstructed haplotypes for haphpipe_assemble_02 pipeline (reference-based assembly); the final consensus sequence for the Geneious (vii) de novo workflow and (vi) reference-based workflow; and finally, the (viii) average between the final two sequences (one for each read file) for HyDRA. The three amplicons are shown for both empirical datasets (HIV: *PRRT*, *int*, *gp120* and HCV: *core*, *E1*, *E2*). There are no results for HyDRA in the *gp120* gene for HIV or for any HCV genes because HyDRA only analyzes the *pol* gene region of HIV.

**Table 1 viruses-12-00758-t001:** Characteristics of the compared programs.

Pipeline	Haphpipe_Assemble_01	Haphpipe_Assemble_02	HyRDA	Geneious	Geneious
**Assembly approach**	De novo	Reference	Reference	Reference	De novo
**Reference**	None	Flexible	HXB2	Flexible	None
**Consensus refinement**	Available	Available	No	Available	No
**Genes**	All	All	*pol*	All	All
**Use**	Free, Command line	Free, Command line	Free, Web-based	Paid GUI	Paid GUI
**Read trimming**	Available, Trimmomatic	Available, Trimmomatic	Yes, parameters set by user	Available, modified-Mott algorithm	Available, modified-Mott algorithm
**Error correction**	Available, SPAdes	Available, SPAdes	Set sequencing platform error rate	Available, BBNorm *	Available, BBNorm *

* BBNorm is part of BBTools package and found at http://seqanswers.com/forums/showthread.php?t=49763.

**Table 2 viruses-12-00758-t002:** Comparison of the de novo and reference-based assembly pipelines in HAPHPIPE for the simulated dataset.

Tool	N	Avg Bowtie2 Alignment First Step	Avg Bowtie2 Alignment Last Step	Number of Samples Refine.02	Number of Samples Refine.03	Number of Samples Refine.04	Number of Samples Refine.05
Simulated HIV data subtype B							
HP01	100	99.87%	99.64%	3	0	0	0
HP02	100	88.95%	95.67%	100	93	2	0
GDN	100	NA	64.20%	NA	NA	NA	NA
GRB	100	NA	82.63%	NA	NA	NA	NA
Simulated HIV data non-subtype B							
HP01	50	98.80%	96.92%	50	0	0	0
HP02	50	59.37%	80.47%	50	44	2	0
GDN	50	NA	64.30%	NA	NA	NA	NA
GRB	50	NA	63.05%	NA	NA	NA	NA

Abbreviations: HP01 = haphpipe_assemble_01 (de novo assembly), HP02 = haphpipe_assemble_02 (reference-based assembly), GDN = Geneious de novo assembly, GRB = Geneious reference-based assembly, N = number of sequences.

**Table 3 viruses-12-00758-t003:** Comparison of the effect of consensus generation on estimated genetic diversity across the empirical datasets.

Tool	Num of Seqs	Avg Bowtie2 Alignment First Step	Avg Bowtie2 Alignment Last Step	Number of Samples Refine.02	Number of Samples Refine.03	Number of Samples Refine.04	Number of Samples Refine.05
***Empirical HIV Data***							
HP01	36	53.67%	44.23%	36	10	0	0
HP02	36	54.69%	47.58%	36	5	0	0
GDN	36	NA	89.98%	NA	NA	NA	NA
GRB	36	NA	27.68%	NA	NA	NA	NA
***Empirical HCV Data***							
HP01	23	90.46%	72.40%	23	19	7	1
HP02	23	78.13%	75.27%	23	22	4	1
GDN *^	23	NA	97.27%	NA	NA	NA	NA
GRB	23	NA	63.31%	NA	NA	NA	NA
***Empirical SARS-CoV-2 Data***							
HP01	4	100%	94.32%	4	3	0	0
HP02	4	100%	94.36%	4	4	0	0
GDN *^	4	NA	80.55%	NA	NA	NA	NA
GRB *	4	NA	94.23%	NA	NA	NA	NA

* Total reads had to be subsampled to 100,000 reads per FASTQ file for Geneious to produce results. ^ Alignment rates are for reads mapped to contigs. Not all contigs were scaffolded to the reference: 27.78% of contigs for HIV, 29.07% of contigs for HCV, and 13.11% of contigs for SARS-CoV-2 data were used after scaffolding. Abbreviations: HP01 = haphpipe_assemble_01 (de novo assembly), HP02 = haphpipe_assemble_02 (reference-based assembly), GDN = Geneious de novo assembly, GRB = Geneious reference-based assembly, N = number of sequences.

**Table 4 viruses-12-00758-t004:** Comparison of the effect of consensus generation on estimated genetic diversity across the empirical HIV dataset.

	*PRRT*					*int*					*gp120*				
Pipeline	N	H	S	π	θ	N	H	S	π	θ	N	H	S	π	θ
HP01	36	7	6	0.0005	0.0008	36	5	5	0.0009	0.0013	36	9	10	0.0012	0.0015
HP02	36	6	6	0.0005	0.0009	36	5	5	0.0009	0.0014	36	10	13	0.0016	0.0016
HP01 haplotypes	132	78	119	0.0028	0.0127	162	52	35	0.0021	0.0064	224	167	149	0.0049	0.0154
HP02 haplotypes	139	85	110	0.0034	0.0123	140	46	34	0.0021	0.0071	70	40	61	0.0026	0.0067
HyDRA	36	1	2	0.0003	0.0003	36	1	2	0.0003	0.0006	NA	NA	NA	NA	NA
GRB	36	1	3	0.0004	0.0005	36	1	3	0.0004	0.0009	36	1	5	0.0006	0.0007
GDN	36	2	4	0.0005	0.0006	36	2	4	0.0007	0.0011	36	3	9	0.0009	0.0011

HyDRA is an average between read1 and read2. Abbreviations: HP01 = haphpipe_assemble_01 (de novo assembly), HP02 = haphpipe_assemble_02 (reference-based assembly), GDN = Geneious de novo assembly, GRB = Geneious reference-based assembly, N = number of sequences, H = number of haplotypes, S = number of polymorphic sites, π = nucleotide diversity, θ = Watterson’s genetic diversity, *PRRT* = protease and reverse transcriptase, *int* = integrase.

**Table 5 viruses-12-00758-t005:** Comparison of the effect of consensus generation on estimated genetic diversity across the empirical HCV dataset.

	*core*					*E1*					*E2*				
Pipeline	N	H	S	π	θ	N	H	S	π	θ	N	H	S	π	θ
HP01	23	16	33	0.0360	0.0379	23	21	188	0.0814	0.0793	23	20	406	0.1047	0.0960
HP02	23	16	26	0.0373	0.0366	23	20	151	0.0703	0.0707	23	23	380	0.1028	0.0960
HP01 haplotypes	63	32	46	0.0328	0.0407	48	36	216	0.0792	0.0739	61	53	462	0.1021	0.0845
HP02 haplotypes	67	30	114	0.0619	0.1241	49	32	228	0.0812	0.0842	56	44	615	0.1213	0.1209
GRB	23	2	377	0.4561	0.3493	23	5	167	0.0736	0.0939	23	3	482	0.1236	0.1484
GDN	23	14	55	0.0826	0.0746	23	16	146	0.0678	0.0691	23	22	389	0.1051	0.0991

Abbreviations: HP01 = haphpipe_assemble_01 (de novo assembly), HP02 = haphpipe_assemble_02 (reference-based assembly), GDN = Geneious de novo assembly, GRB = Geneious reference-based assembly, N = number of sequences, H = number of haplotypes, S = number of polymorphic sites, π = nucleotide diversity, θ = Watterson’s genetic diversity.

**Table 6 viruses-12-00758-t006:** Comparison of the effect of consensus generation on estimated genetic diversity and adjusted genetic p-distance to the reference sequence across the empirical SARS-CoV-2 dataset.

	Diversity Estimates					Adjusted Genetic p-Distance		
Pipeline	N	H	S	π	θ		Average	STDEV
**HP01**	4	1	0	0	0	**HP01 Initial**	0.0033	0.0035
**HP02**	4	1	0	0	0	**HP01 Final**	0.0035	0.0034
**GDN**	4	4	371	0.0081	0.0083	**HP02 Initial**	0.0019	0.0015
**GRB**	4	1	13	0.0004	0.0004	**HP02 Final**	0.0021	0.0019
						**GDN**	0.0417	0.0423
						**GRB**	0.0031	0.0008

Abbreviations: HP01 = haphpipe_assemble_01 (de novo assembly), HP02 = haphpipe_assemble_02 (reference-based assembly), GDN = Geneious de novo assembly, GRB = Geneious reference-based assembly, N = number of sequences, H = number of haplotypes, S = number of polymorphic sites, π = nucleotide diversity, θ = Watterson’s genetic diversity, STDEV = standard deviation.

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
