# Peer review of "Validation of Variant Assembly Using HAPHPIPE with Next-Generation Sequence Data from Viruses"

_viruses, 2020, doi:10.3390/v12070758_

Round 1

Reviewer 1 Report

This highly

This is an interesting, well-designed and well-executed study. All parts of this article are clear, reasonable and comprehensive. 

There is a need for the use of next-generation sequencing (NGS) methods to identify and study low abundance intra-host viral sequences in the study of viral diseases. Although the body of new data coming through NGS methods is growing there is a lack of technical tools allowing quick and comprehensive analyses of these big data including reference-based assembly of viral NGS data. The manuscript by Gibson et al. is an interesting paper describing a new open-source tool enabling reference-based assembly of NGS data. This is strictly a technical paper describing in detail how this new software can be run while comparing it to the similar software already available. The paper is written clearly with detailed description of how new users can run it (with multiple examples described). It includes all the details and is written in proper English. The paper is long in its current form and it would benefit if shortened.

Author Response

We would like to thank the review for their comments. The paper has been amended by removing redundant segments. We believe the shortened version has increased accessibility for readers with all levels of bioinformatics experience. 

Reviewer 2 Report

The manuscript presents a bioinformatics pipeline for determining reference sequences as well as calling intra-patient haplotypes from virus NGS data. The method is shown to be more effective than two common GUI based programs and accurate when the true sequence is known (from simulated data).

The biggest issue I have with this work is that it is tested against GUI based programs while itself not being GUI based. Thus, likely the primary users will be those already not using GUI based programs unless the authors can truly make this more assessible. Thus, while it is pointed out that beneficial additional work would be validated against the existing bioinformatic programs it seems that should have been the primary goal here. In absence of this I think there needs to be additional focus and build out of making HAPHPIPE be truly intuitive and able to be used by anyone with very basic bioinformatic training.

            In actually running this program, I agree the guides and details provide make it much easier than most programs, however, I am not convinced it is yet to the point that users with limited bioinformatic knowledge could use it without much trouble. The biggest shortcoming in running the pipeline is the need for two additional outside programs- particularly with PredictHaplo. I encounter multiple permission errors on my server and would not have easily been able to install this without bioinformatics knowledge (getting it installed took me longer than any other part of the pipeline).

In keeping with the above comments some suggestions for instructions document (non-published material):

Recapitulate everything you need to install PredictHaplo. Allowing users to completely install and run this program without finding any additional websites is one step closer to allowing this to be done by people with minimal bioinformatic experience.

            Note: I now find this on the website- which covers almost all edits I was going to provide on this document. I am unsure what the difference between the non-published document and the website are supposed to be? The website contained some additional information and lines of code needed to install/run programs but the provided document was simply written and well explained. I recommend merging that document into the website. For example, see STACKS- this program revolutionized molecular ecology and evolutionary studies not by being the best program but by having incredibly easy to follow user instructions and being a great bridge into bioinformatics. https://catchenlab.life.illinois.edu/stacks/ The HAPHPIPE website already started merged with the document provided with this paper I think could accomplish these goals with viral NGS.

Back to manuscript:

An additional feature that I think could be worked in is the ability to have the haplotypes coded for synonymous and non-synonymous changes have an output table of amino acid changes and the frequency of them within a patient. For example, see the iVar program that does this nicely for the tiled amplicon NGS approach. The outputs from this program are very close to this stage- and easy for an experienced user to generate. However, this would provide an additional piece to this pipeline that makes things more intuitive for the inexperienced user.

Line 812: Please provide an additional sentence or two discussing the haplotype callers. For example, they do the worse with higher diversity viruses such as HIV and that PredictHaplotype was found to be the best out of the twelve evaluated. This seems to be one of the major advantages of HAPHPIPE over V-pipe.  

As a note, I think the name really matters in how much these protocols take off. It took me a second to get this one, but HAPHPIPE pronounced halfpipe is excellent! This was further solidified by your skateboarding virus!

Author Response

The manuscript presents a bioinformatics pipeline for determining reference sequences as well as calling intra-patient haplotypes from virus NGS data. The method is shown to be more effective than two common GUI based programs and accurate when the true sequence is known (from simulated data).

The biggest issue I have with this work is that it is tested against GUI based programs while itself not being GUI based. Thus, likely the primary users will be those already not using GUI based programs unless the authors can truly make this more assessible. Thus, while it is pointed out that beneficial additional work would be validated against the existing bioinformatic programs it seems that should have been the primary goal here. In absence of this I think there needs to be additional focus and build out of making HAPHPIPE be truly intuitive and able to be used by anyone with very basic bioinformatic training.

We thank the Reviewer for their suggestion and understand their concerns. However, the primary focus of this validation study was to assess HAPHPIPE’s performance in a clinical and/or public health setting. For this reason, we chose to compare HAPHPIPE against HyDRA and Geneious. The former is popular among the HIV research community and the latter is a representative standard of commercial software frequently used in genomics analysis. Thus, the primary users would in fact be likely to use these programs. Additionally, as HAPHPIPE was created with users of limited bioinformatics background in mind, we believe HyDRA and Geneious are fitting candidates for comparison as their GUI based interface makes them accessible to novice users. 

         In actually running this program, I agree the guides and details provide make it much easier than most programs, however, I am not convinced it is yet to the point that users with limited bioinformatic knowledge could use it without much trouble. The biggest shortcoming in running the pipeline is the need for two additional outside programs- particularly with PredictHaplo. I encounter multiple permission errors on my server and would not have easily been able to install this without bioinformatics knowledge (getting it installed took me longer than any other part of the pipeline).

We understand your concern. In the case that a researcher new to bioinformatics encounters difficulties with PredictHaplo, we have added a second haplotype program option: CliqueSNV. CliqueSNV is easier to install and performed second best to PredictHaplo in a haplotype validation study (doi: 10.1016/j.meegid.2020.104277). Additionally, we have spent a significant amount of time increasing the approachability of the User Guide for users new to using the terminal. We had new users (including beginning undergraduate students) help use, create, and debug a more approachable User Guide, as well as use and debug the program itself. Because of these measures, we feel that a scientist new to bioinformatics could independently install and execute this program with their own data, especially now with the inclusion of CliqueSNV, which is easier to obtain for a new user. 

In keeping with the above comments some suggestions for instructions document (non-published material):

Recapitulate everything you need to install PredictHaplo. Allowing users to completely install and run this program without finding any additional websites is one step closer to allowing this to be done by people with minimal bioinformatic experience.

         Note: I now find this on the website- which covers almost all edits I was going to provide on this document. I am unsure what the difference between the non-published document and the website are supposed to be? The website contained some additional information and lines of code needed to install/run programs but the provided document was simply written and well explained. I recommend merging that document into the website. For example, see STACKS- this program revolutionized molecular ecology and evolutionary studies not by being the best program but by having incredibly easy to follow user instructions and being a great bridge into bioinformatics. https://catchenlab.life.illinois.edu/stacks/ The HAPHPIPE website already started merged with the document provided with this paper I think could accomplish these goals with viral NGS.

Thank you for your review of our unpublished document. We are glad you found it thorough and concise. We agree that merging the document and the User Guide would be beneficial. The paper is going through review currently and will be linked to the User Guide upon publication (which will be open access).

As for the comment about PredictHaplo, we have included a second option for a haplotype program: CliqueSNV. CliqueSNV is very easy to install and performed second best to PredictHaplo in a haplotype validation study (doi: 10.1016/j.meegid.2020.104277). Therefore, we believe that the inclusion of this module will prove accessible and adequate for those with minimal bioinformatics experience. 

Back to manuscript:

An additional feature that I think could be worked in is the ability to have the haplotypes coded for synonymous and non-synonymous changes have an output table of amino acid changes and the frequency of them within a patient. For example, see the iVar program that does this nicely for the tiled amplicon NGS approach. The outputs from this program are very close to this stage- and easy for an experienced user to generate. However, this would provide an additional piece to this pipeline that makes things more intuitive for the inexperienced user.

We agree that such a feature would be a valuable addition and thank the Reviewer for the suggestion. However, the focus of this paper is a validation study for the purpose of assessing HAPHPIPE’s performance against current industry standards, not a manuscript regarding the pipeline itself (as is the unpublished document attached). We will consider this as another module to be implemented in future versions of HAPHPIPE.

Line 812: Please provide an additional sentence or two discussing the haplotype callers. For example, they do the worse with higher diversity viruses such as HIV and that PredictHaplotype was found to be the best out of the twelve evaluated. This seems to be one of the major advantages of HAPHPIPE over V-pipe.  

Information regarding why PredictHaplo was chosen are included in the manuscript in lines 167-169 (“PredictHaplo was chosen for the HAPHPIPE suite because it was determined to have the best performance for capturing intra-host viral variation compared to eleven other haplotype reconstruction tools in a recent study [21] of diversity levels observed in viral intra-patient data.”) and in relation to V-pipe in lines 811-813 (“V-pipe implements HaploClique and Savage for global haplotype reconstruction and ShoRAH for local reconstruction, all of which performed poorly relative to PredictHaplo in a recent comparison of haplotype reconstruction tools”). 

As a note, I think the name really matters in how much these protocols take off. It took me a second to get this one, but HAPHPIPE pronounced halfpipe is excellent! This was further solidified by your skateboarding virus!

Thank you for enthusiasm about the name!